# Considerations in Planning Physical Activity for Older Adults in Hot Climates: A Narrative Review

**DOI:** 10.3390/ijerph18031331

**Published:** 2021-02-02

**Authors:** Lydia See, Rohan L. Rasiah, Rachael Laing, Sandra C. Thompson

**Affiliations:** 1Western Australian Centre for Rural Health, School of Population and Global Health, University of Western Australia, Crawley 6009, Australia; rohan.rasiah@uwa.edu.au (R.L.R.); rachaellaing96@gmail.com (R.L.); sandra.thompson@uwa.edu.au (S.C.T.); 2Oral Health Centre of Western Australia, School of Dentistry, University of Western Australia, Nedlands 6009, Australia

**Keywords:** exercise, ageing, heat, ambient environment, temperature, elderly, thermoregulation, fitness intervention

## Abstract

Regular physical activity has multiple health benefits for both the prevention and management of disease, including for older adults. However, additional precautions are needed with ageing given physiological changes and the increasing prevalence of comorbidities. Hot ambient temperatures increase the risks of exercise at any age, but are particularly important given thermoregulatory changes in older people. This narrative review informs planning of physical activity programs for older people living in rural areas with very hot climates for a period of the year. A multi-database search of peer-reviewed literature was undertaken with attention to its relevance to Australia, starting with definitions and standard advice in relation to physical activity programming and the incremental limitations imposed by age, rurality, and extreme heat. The enablers of and barriers to increasing physical activities in older adults and how they can be modified for those living in extreme hot climates is described. We describe multiple considerations in program design to improve safety, adherence and sustaining physical activity, including supervision, simple instructions, provision of reminders, social support, encouraging self-efficacy. Group-based activities may be preferred by some and can accommodate special populations, cultural considerations. Risk management is an important consideration and recommendations are provided to assist program planning.

## 1. Introduction

The health benefits of physical activity are increasingly recognised and extolled to the population for preventing or delaying the onset of disease [1]. They are also promoted as improving outcomes for medical conditions, when only decades ago, bed rest may have been prescribed. The evidence supporting benefits of physical activity extends across multiple conditions, and includes cardiovascular health, reducing risks for chronic disease, weight loss, benefits for muscles, bones and the brain, happiness, sleep quality, mental health, and prevention of dementia [2]. Exercise is now prescribed for people with a number of conditions, such as diabetes, cancer and heart disease, so safe and appropriate exercise advice is needed for older people where the prevalence of these conditions is higher, through an activity plan for achieving recommended physical activity that integrates preventive and therapeutic recommendations [3]. For the purposes of consistency, this paper will use physical activity as our key term and focus of interest, with the understanding that exercise is a subcategory.

Associated with improvements in standards of living and medical care has been increasing longevity and an increasing proportion of older people in the population. For example, Australia has one of the highest life expectancies in the world, increasing from 71.3 years in males and 78.3 years in females in 1980–1982 to 80.5 and 84.5 years in Australian males and females respectively in 2014–2016 [4]. Over the 20 years from 1999 to 2019, the 65 years and older cohort has increased from 12.3% to 15.9% [5]. This means that people are living longer and are encouraged to keep physically active at older ages. Despite the advice to maintain physical activity across one’s lifespan, an estimated three-quarters of Australians over the age of 65 years old are not achieving the recommended amount of weekly physical activity [6]. The same phenomenon of demographic ageing with increase in life expectancy is occurring in both developed and developing countries [7], with efforts to reduce premature disability through health behaviour change, that includes exercise at older ages, common in many countries [8].

Ageing is stochastic and heterogeneous, with individuals experiencing a different ageing trajectory. Hence, physical activity advice for older people must be adjusted considering the range of individual ageing and environmental factors affecting one’s ability, confidence and motivation to be physically active. Active older adults have been shown to have lower rates of all-cause mortality, coronary heart disease, high blood pressure, stroke, type 2 diabetes, colon cancer, breast cancer, a higher level of cardiorespiratory and muscular fitness, healthier body mass and composition and a biomarker profile that is more favourable for the prevention of cardiovascular disease, type 2 diabetes and the enhancement of bone health [1,2,9]. However, the physiology of ageing along with concurrent development of non-communicable diseases may prevent an individual from leading an active lifestyle, although coexisting chronic disease should not be a deterrent in undertaking physical activity. In fact, multiple guidelines from the Australian government and WHO indicate that individuals unable to undertake the recommended guidelines can still be active to the level their abilities and health conditions allow and work to incrementally increase the length of time they undertake activity towards at least the minimum recommended guidelines [9,10,11]. The current World Health Organisation (WHO) recommendations for physical activity for older adults are 75–150 min of vigorous-intensity and 150–300 min of moderate intensity aerobic physical activity per week [12]. Muscle strengthening activities at moderate or greater intensity involving all major muscle groups should be carried out on two or more days a week [12]. Additionally, varied multicomponent physical activity emphasising functional balance and strength training at moderate or greater intensity should be undertaken three or more days a week. Overall, older adults should also aim to reduce sedentary behaviour and reduce sedentary time [12].

Despite these recommendations, many developed countries have documented poor physical activity levels as people age. In Australia, 69% of men and 75% of women aged over 65 years were insufficiently active [13]. This was comparable with Scotland, where only 31% of men and 21% of women over 75 years meet their recommendations [14]. In England, the proportion of inactive older adults is at 27% for those 75–84 years and increases to 75% in those 85+ [14]. In America, the prevalence of inactivity is reported to be lower at 27% for 65–74-year-olds and increases to over 35% for those over 74 years [14]. Although the data reported here cannot be compared with each other, it is evident that there is a substantial older adult population that have insufficient physical activity. Therefore, considering the population trend of increasing older age, it is important to design and implement physical activity programs that target this population and improve the uptake of recommendations.

In Australia, many existing physical activity programs for older adults have been developed in metropolitan areas which are largely characterised by a temperate climate. However, there are many places in Australia (especially Northern Australia) where the daytime temperature is regularly above 35 °C for many months a year. This inevitably means that programs need to be designed with the potential vulnerabilities of older adults in mind and recognising that the environment constrains some of the more standard approaches to exercising at an older age. The importance of hot climates is that heat production arises from muscular activity and is a function of aerobic activity and the exercise intensity, which must be balanced by the transfer of heat from the body to the environment, which involves a number of mechanisms, including evaporative capacity, which are constrained by high ambient temperatures [15].

This narrative review was undertaken to inform planning for a physical activity program to be delivered in a regional city in a northern region of Australia. Our interest as academics and practitioners was to promote physical activity in older adults living in this remote location, where there are extremely high ambient external temperatures. As similar conditions occur across northern Australia and in many other areas of the world, the review has much broader relevance. Moreover, global warming means that extreme weather events and heat stress are likely to increase [16]. The review discusses current knowledge related to physical activity in older people, and considerations for programming in rural areas, hot climates and with underserved communities. This information can inform best practice approaches for the planning and delivery of physical activity to older people in similar circumstances, and appropriate strategies for those tasked with delivering health promotion and physical activity programs, given that most training is based in metropolitan population centres in more temperate climates.

## 2. Methods

A narrative review provides an overview of issues and an evidence round-up on a specific health care topic, with advantages and disadvantages compared to systematic evidence-based reviews [17,18]. It does not answer one specific research question, instead providing information around a topic of interest. Narrative reviews are most useful for obtaining a broad perspective on a topic by bringing together and discussing the state of the science on a specific topic or theme from a theoretical and contextual point of view. There is a recognised important place for narrative reviews in deepening understanding [19]. This review was undertaken in 2020 and draws upon international peer-reviewed literature, starting with definitions and standard advice in relation to physical activity programming and exploring the additional incremental limitations imposed by age, rurality, and extreme heat. We also explore the enablers of, and barriers to, increasing physical activities in older adults and how they can be modified for those living in extreme hot climates.

### Search Strategy

After an initial search for relevant information on older people, exercise/physical activity and hot climates, we identified salient issues to be explored and that there was a lack of relevant reviews on this issue. Given that the intention of a narrative review is to describe and synthesize the available literature on a topic, we agreed on broad, relevant search terms and a structure for the review. In keeping with the approaches described by Gussenbauer and Haddaway, we utilised a combination of look-up and exploratory search strategies to allow for discovery and learning, fact checking and navigation to new relevant areas for the review that emerged during the search for authoritative knowledge sources [20]. The iterative process of searching extended across the months of writing and refining the review. Peer-reviewed literature was searched using PubMed, CINAHL and Google Scholar databases. The search strategy used PubMed: “Older Adults” AND “Hot OR tropical” AND “climate OR weather” AND “physical activ+”. Within CINAHL, MeSH terms and search combinations were used that involved the keywords “Physical activity”, “older adults”, “barriers or obstacles or challenges or difficulties or issues or problems”, “rural and remote”, “Australia, Australians”. A grey literature search was also undertaken using Google Scholar as the search engine, along with specific searches of the Australian government and World Health Organisation websites. This was to ensure that existing guidelines from local and international bodies along with existing literature in the form of articles are incorporated into our recommendations of the considerations required in planning physical activity for older adults in hot climates. Ongoing discussion occurred within the research team regarding how much background and detail should be included, given that the area of the review reflected the intersection of three distinct areas and refinement of the review and presentation of material occurred throughout the writing process.

## 3. Definitions

### 3.1. Older Adults

What is meant by an older population varies in different contexts, with the potential to be defined by chronological, medical and cultural means [21]. The World Health Organisation (WHO) refers to older adults as being 60 years and older [8], whereas in Australia, the term of older adults refers to those aged 65 years and older [5,21]. Ultimately, the term is used to reflect biological and functional ageing in which physical and cognitive effects align to constrain participation in activities that would generally have been readily accomplished at a younger age. Some populations, such as Aboriginal and Torres Strait Islander Australians (among other First Nations peoples), experience a disproportionate disadvantage and thus have a reduced life expectancy and experience a higher rate of morbidities at an earlier age compared to their less disadvantaged counterparts. Thus, in Australia, Aboriginal and Torres Strait Islanders are considered eligible for aged care services from the age of 50 years [10], compared to the standard 65 years and older in the general Australian population [22,23].

### 3.2. What Is “Healthy Ageing”?

Healthy ageing is defined “*as the process of developing and maintaining the functional ability that enables wellbeing in older age”* [24,25]. Functional ability is a combination of intrinsic capacity of an individual (a combination of physical and mental capacities) and “*the environment in which he or she lives and the interactions among them”* [24]. Rowe and Kahn (1997) defined successful ageing as involving three components of freedom from disease and disability, high cognitive and physical functioning and active engagement with life [10,26]. Function has been identified as a key concept in both healthy and successful ageing, however it should be noted that different cultures will value ageing and the aged differently. Therefore, the value of physical activity in older adults will also vary across cultures and so may be particularly impacted by environmental factors such as hot climates.

### 3.3. Definitions of Physical Activity

Physical Activity is “*any bodily movement produced by skeletal muscles that requires energy expenditure and produces progressive health benefits”* [10,27] and includes exercise and bodily movements from playing, working, active transportation, house chores and recreational activities [27]. Incidental physical activity refers to the movements that occur in carrying out activities of daily living [10].

Physical activity is beneficial no matter the type [27]. Traditional advice based on evidence is that to produce health benefits, individuals should undertake 150 min or more of moderate and/or vigorous activity per week [9,10]. Moderate physical activity is defined as any *“activity undertaken at a level that causes the heart to beat faster and some shortness of breath, but during which a person can still talk comfortably”*(p. 7) [10]. Vigorous physical activity refers to any activity that *“causes the heart to beat a lot faster and shortness of breath that makes talking difficult between deep breaths”*(p. 8) [10].

### 3.4. Types of Exercise

Exercise is defined as a subcategory of physical activity that is “planned, structured, repetitive and purposeful in the sense that that improvement or maintenance of one or more components of physical fitness is the objective” [10,27]. It can be divided into four main categories of endurance, strength, flexibility, and balance; each category has its own recommended duration, frequency and type of exercise regimen (Table 1).

A physical activity program developed for older persons should ensure that four domains of physical literacy are included to help maintain sustained physical activity. In practice, this means that any program for older persons needs to include social interaction, movement, enjoyment and modifications for age and environment [29]. Given multiple differences between older adults, a “one-size-fits-all” approach is unlikely to work. Therefore, program designs need to be customised based on the psychological, physical, cognitive and social circumstances of the target groups. This means using a multi-level approach, focussing on individuals and considering interpersonal and intrapersonal factors, as well as how opportunities are impacted by organisations, the community and policies [29].

## 4. Factors Affecting Physical Activity in Older Adults

### 4.1. Physiological Changes, Thermoregulation, and Comorbidities in Older Adults

Ageing is characterized by a progressive decline in the physiological reserve of all organ systems. This occurs at different rates and varies in different individuals [30]. Most organ systems show a physiological reduction in function with age, accompanied by progressively reduced redundancy of function and ability for repair. Even as people become frailer with age, there are benefits of maintaining physical activity [31].

Thermoregulation refers to the ability of the human body to maintain a core body temperature at around 37 °C, within a narrow range of 35–39 °C [32], despite exposure to a wide range of external temperatures. Thermoregulation is critical for human survival and cellular functions. Effective maintenance of core body temperature is based on several homeostatic mechanisms. Internal heat is produced by cellular respiration and this needs to be balanced by heat loss from the skin surface to the surrounding environment through dry and evaporative heat exchange [32]. Heat stress can occur either with high environmental temperatures or with physical activity in which 80% of muscle contraction is produced as heat. There are two mechanisms which allow heat to be lost to the environment, namely increased sweating and skin blood flow [32]. Evaporation of sweat allows the greatest capacity for heat loss during environmental heat exposure and physical activity [33].

Decreased heat tolerance and alterations in the ability to thermoregulate occur with ageing [34,35]. Those aged 60 years and older are particularly badly affected by extreme heat, reflecting physiological changes of ageing that impact thermoregulation and the accumulation of comorbidities that increase their vulnerability [35,36]. For those involved in designing exercise programs for older people in regions of high heat, it is important to understand why this is to inform how these age-related affects need to be accommodated in programs and be overcome.

Thermoregulation is closely related to physical fitness, and fitness decreases with advancing age [35]. In older adults, multiple studies have shown that the threshold for sweating in older ages may have a delayed onset as core temperature rises and overall sweat rates are lower, so there is a greater potential for heat to be stored within the body, causing core body temperatures to rise to dangerous levels [32]. In general, heat exposures cause an increase in skin blood flow [37], but older adults have less ability to adjust skin blood flow as a result of changes to core temperatures. Furthermore, too much increase in skin blood flow means that blood pressure needs to increase, and increased cardiac output during physical activity is also necessary to maintain skeletal muscle perfusion. For older adults, a higher cardiac output may put pressure on the cardiovascular system, a concern for those who have compromised cardiovascular function and exacerbated with poor hydration [38].

Sweating rate decline in the elderly has been found to depend upon the environment. In hot dry environments, older persons consistently secrete sweat at a lower rate than their younger counterparts, although this difference tends to disappear as the humidity of the environment increases. Hydration, both in terms of body water content and compartmentalisation and of skin water content, is a key issue, particularly in the elderly. Older adults may have a decreased ability to sense and adapt to dehydration [34].

Regular exercise has many health benefits across a range of diseases, including in older people. Physical activity training slows the known age-related decline in sweat gland output in fit older adults compared to unfit older adults [32]. Additionally, mechanisms that enable heat loss, namely skin blood flow and sweating, improve with both aerobic and resistance exercise training [39]. Thus, exercise training appears to improve thermoregulatory control by increasing heat-induced skin blood flow and sweating responses, allowing for better internal heat distribution and evaporative heat loss capacity of older adults [32]. Regular exposure to high heat may also reduce age-related decline in vascular endothelial function [40], and help maintain thermoregulatory functions through physiological adaptation [41,42].

Across the world, people are now generally living longer, reflecting the benefits of modern science and medicine [43]. However, even if their quality and length of life is maintained, there is an increase in the number of health conditions that they live with. Increasingly, we recognise that older people often live with two or more chronic conditions, meeting the definition of multimorbidity [44]. Multimorbidity is associated with decreased quality of life, functional impairment, increased health care utilization, polypharmacy, increased workload for self-management and increased mortality [45]. Associated with managing these diseases is the use of medications to help control the conditions, and these may have their own physiological effects.

Common chronic diseases in older adults in developed countries are hypertension, diabetes mellitus, cardiovascular and respiratory disease, cancer and obesity. Exercise is recommended in all these diseases as part of their management. However, the risk of heat-related illness during high heat conditions is greater in some of these diseases because of physiological impairment in temperature regulation in hot conditions, and is the subject of a recent review to which the reader is referred for more detailed information and disease-specific information on obesity, diabetes, cardiovascular and respiratory disease [34]. Different mechanisms operate for increasing risk for older people exercising in health with different diseases. For example, obesity may interfere with heat-sensing (small fibre neuropathies) and heat-dissipating abilities (lower surface area to body mass for evaporation), lower thermal conductivity of fatty tissue, restrictive conductive loss transfer and the extra weight increases the metabolic costs of weight-bearing activities, elevating the rate of heat production [34]. Other diseases such as cancer and osteoarthritis also increase with age. There is now evidence that exercise is generally helpful in preventing and managing these conditions, improving symptom management and increasing survival [46,47,48,49,50]. Physical limitations may compound the usual physiological effects of ageing and heat and require additional consideration in planning exercise programs.

The thermoregulatory changes associated with ageing require those planning physical activity for older adults in high temperature environments to recognise potential risks and adapt programs. Recommendations regarding levels of safe physical activity specific to older adults in heat (for example, in summer or areas with very hot weather) will increasingly be needed with climate change, associated temperature increases and awareness of the need for safety in testing the body’s ability to adapt to such changes.

### 4.2. Barriers and Enablers

To encourage older adults to habitually engage in regular physical activity, a range of factors need to be considered in the design of any physical activity program. Better adherence to physical activity occurs when programs utilise extrinsic motivators (e.g., cash incentives and smartphone applications), are supervised, incorporate simple instructions, provide reminders, offer social support and reinforcement and encourage self-efficacy [51,52,53]. Additionally, factors such as socioeconomic status, educational levels, marital status, general and mental health status, physical ability, and cognitive ability can influence the level of physical activity [51]. Multiple factors impact the adherence of older people in an exercise program: their beliefs about exercise and understanding of the benefits of exercise, past experiences with exercise; their goals, personality and any unpleasant sensations associated with exercise [54,55]. Brief advice by a healthcare professional is an effective motivator [56,57].

Much like physical literacy, the considerations for and enablers of physical activity are multilevel and rely on engagement at the individual, community and organisational levels [58] and their associated personal, social, environmental and structural barriers and enablers [58]. Using the social–ecological framework, relevant factors (Table 2) can be classified as intrapersonal, interpersonal, physical environment and structural and organisational. Intrapersonal factors are those related to the individual such as physical and/or mental health and individual preferences, whereas interpersonal factors pertain more to group activities and social interactions as being the motivators and enablers to engage older adults in physical activity [58,59]. The physical environment includes the built environment and factors such as convenience of access, safety of neighbourhoods, quality of walkways and parks, as well as weather. All these factors can enable or limit physical activity amongst older adults. Structural and organizational factors incorporate considerations of available facilities and physical activity programs, including costs, flexibility, quality instructors, considerations of appropriateness of activities to different fitness levels and physical limitations, which can all influence the physical activity levels of older adults [58].

The type of exercise and perceived benefits influences the uptake of programs, with Burton et al. reporting that motivators to participating in resistance training included preventing deterioration (disability), reducing risk of falls, building (toning) muscles, feeling more alert, and better concentration [60]. Concerns about looking too muscular or thinking participation increased the risk of having a heart attack, stroke, or death, which were barriers to participation. Program participants have also reported that enjoying being with others while exercising and desiring a routine that promoted accountability, with environmental motivators including marketing materials, encouragement from a trusted person, lack of program fees, and the location of the program [61]. Boehm and colleagues explored literature related to older adults in rural and remote areas of Australia to inform research into falls prevention and identified 20 different barriers to exercise and 14 different facilitators.

There are numerous factors that increase the likelihood of physical inactivity, some of which are not able to be modified (for example, biological factors such as age, sex and race), while others are able to be modified. Modifiability relates to whether changes can occur to allow for increased physical activity, where engagement at a personal or broader socioecological level can make a difference in the short or long term to addressing inactivity in older adults (Table 2). The ease in which this occurs depends on an individual’s circumstances. Potential modifiable factors are considered as those where change is possible but would take a longer time and may relate to an individual’s behaviour or motivation, but can also include factors which rely on community, organisations, or government policies for changes to the environment which favour physical activity. It is important to review these inactivity factors when determining physical activity programs. The ability to motivate an individual to engage in any activities will be determined by how effective a program is in overcoming such factors.

### 4.3. Environmental Factors

The environment in which older people live contributes significantly to their ability to undertake physical activity [59,62,63,64,65]. These factors contribute to a varying extent to community members being active. Many relevant factors can be modified, although there can be significant economic costs to individuals and government [66].

#### 4.3.1. Physical Environment

The physical environment includes both the landscape and natural environment that surrounds a community, for example, soil/land, climate and water supply and the built environment. All of these can affect both participation and the level of physical activity undertaken by older people. Safe, walkable, and aesthetically pleasing neighbourhoods, with access to overall and specific destinations and services, are known to positively influence older adults’ physical activity participation [62]. Local infrastructure has an important influence on physical activity, and while most research has been undertaken in larger communities, rural and remote communities have similar and extra attributes that may impact engagement with physical activity. When considering the physical environment for promoting physical activity for older adults, attributes that are important include the quality of walking paths (e.g., concrete, smooth concrete pavement, gravel or rocky/unstable); gradient of the paths; their accessibility and location from home; lighting; surrounds (for example business, residential or bush setting); perceived safety and security of the area; traffic; and the aesthetics and cleanliness of the space [59,67,68]. Unsurprisingly, older adults will make their own risk assessment and avoid those environments seen as risking falls and/or their physical safety. Easily accessible and low-cost recreational facilities that provide age-appropriate group activities are preferred by older adults and air-conditioned indoor gyms and pools may be particularly favoured in high temperatures [69,70]. One review highlighted that one specific recreational facility appears to be insufficient and a mixture of age-friendly recreational facilities which could include parks and open spaces is preferable to allow older adults to engage in multiple potential forms of physical activities/sports, such as walking, cycling and other sports (tai chi, dancing, etc.) [71]. For the rural context, the availability of a variety of safe environments and facilities is subject to the population needs and interests, and the financial contributions of the government and/or private sector to help with providing facilities and spaces to undertake physical activity [72]. The connection between homes and facilities is important since ease and safety of access are significant factors in older adults undertaking physical activity (both organized and unstructured) in their leisure time. Walking and cycling trails around a community enable opportunities for community members to use. However, their design affects the usability of the trails by older adults, particularly if the trail consists of areas that are uneven, high gradient or poorly lit, as may occur in bush/outback trails [70].

#### 4.3.2. Timing of Exercise to Accommodate Climatic Factors

In a hot climate, and in order to avoid the extremes of heat and ultraviolet (UV) radiation, either the type of exercise, the timing or modifying the immediate internal environment are important considerations for both individual and group activities [73,74], particularly for older people. Swimming or activities where there is good air conditioning will still be possible even when the outside temperature is very high. These areas are usually in gyms and facilities that charge fees for their use, with costs serving as an impediment to use for some. While pools provide opportunities for older people to undertake low impact exercise, some participants may be self-conscious, resulting in a reluctance to using the facilities. Older people may be more conscious of the need to avoid UV exposure because of skin conditions or medications and this may reduce their use of outdoor pools in some rural communities. One approach is to timetable outdoor physical activities such as walking, cycling, and outdoor sports for when temperatures are lower (early morning and late evening). Timing at the end of the day may not be convenient for some, so solutions include the use of air-conditioned spaces and swimming pools.

### 4.4. Special Population Groups

All countries have population subgroups that are particularly disadvantaged and have poorer health. Just as a nuanced understanding is needed for physical activity programs for older people and for people living in hot climates, so must programs be adapted to meet the needs of minority or special subpopulation groups.

Worldwide, indigenous people are among the most marginalised and disadvantaged populations in terms of their social, economic and health status. The political history of Indigenous dispossession, violence and social exclusion has significant bearing on the state of their health. While it is beyond the scope of this review to examine in-depth special considerations for programs for older Indigenous people, they represent a higher proportion of the population living in northern Australia than elsewhere in Australia. Hence, we briefly describe some special considerations from the literature that need to be considered when designing programs for this special population group.

Dahlberg and colleagues examined qualitative studies that had explored issues related to the perspectives of indigenous Australians around physical activity [75]. A key factor affecting the success of a physical activity program is the involvement of users in the design of the program, particularly among indigenous Australians, and three of the four themes identified are relevant for older indigenous people. Preference was given for physical activity in the context of family, group and community activities. There were gender differences in preferences, with indigenous women preferring family-focused activities, whereas men highlighted sport and going on walkabout. The place and nature of the activity was considered important. These qualitative findings have been confirmed by the evaluation of programs undertaken with indigenous Australians with community-focused and group-based programs found to be more effective than individualized programs [76,77,78,79,80]. Group-based programs afford a range of social benefits which positively influence participation in physical activity, helping to create a sense of connection and community among participants [75]. Hence, many successful physical activity programs extend beyond group fitness training to include gym sessions, cooking workshops and (gender-specific) yarning sessions, to enable the reach and retention of participants [75]. Furthermore, a flexible program that allows group participants to meet in a supportive and culturally secure environment positively influences attendance and adherence [76]. Physical activity programs having a holistic and community focus can have advantages that go beyond increasing physical activity. For example, Passmore and colleague’s evaluation of an Australian state-wide physical activity program targeting indigenous Australians found that the approach they utilised improved not only physical activity levels, but also knowledge of nutrition and physical activity, increased community pride and awareness, and better awareness of healthy lifestyles in the community [78]. This may reflect the need for more holistic approaches and that exercise programs with older indigenous Australians have often been conducted in the context of managing chronic diseases where participants have multiple risk factors [81,82].

Socialisation and the promotion of health benefits are also considered important for attracting culturally and linguistically diverse (CALD) groups to exercise programs [83]. Of note, Pasick and colleagues recommend adopting general health promotion principles in promoting exercise to CALD communities by avoiding targeting an intervention to a community characterized by racial or ethnic designation [84]. Instead, they recommend identifying the attributes related to health behaviour, not just ethnic background, defining groups by attitudes, beliefs, cultural concepts and cultural dimensions to health practices. While activities could be tailored by culture as necessary, when appropriate, it was recommended to reach across cultures [84]. This advice is particularly pertinent in rural and remote areas where resources may be limited; hence, holistic and inclusive approaches are particularly important.

### 4.5. Risk Management

It is important to recognize that while there are risks associated with physical activity, there are well documented and substantial benefits of activity in older people that are likely much greater than any health risks arising from physical inactivity [2,10]. In fact, physical activity is prescribed and shown as beneficial for a wide range of health problems, including in older people [85,86]. The advice of encouraging people to gradually increase their physical activity, starting with low intensity and progressing to moderate intensity is particularly relevant for older people, with those who have engaged in physical activity across their lifetime able to continue at the higher level of [10]. Breaks in exercise routines to allow for recovery from aerobic activity and to drink water are of particularly importance in older people and in hot ambient climates. Watching people to ensure they are in fact drinking might be an extra precaution needed when the temperatures are very hot, particularly if they are breathing rapidly and perspiring. In older people, it is particularly important to stress to participants to start their exercising slowly and not overdo it, and to stop the activity at least temporarily if they have symptoms such as chest pain, difficulty breathing, moderate pain, or feel dizzy [10].

There are some conditions that require special consideration in older people, particularly the risk of falls and concerns about cardiovascular disease. Chronic diseases that increase with age, such as heart failure, arthritis, lower respiratory reserve, and diabetes are all likely to require some modification to exercises [32,34].

Pre-exercise screening and assessment is recommended; depending upon the age of participants and existing medical conditions, a medical assessment and referral may be indicated [87]. However, there is concern that requiring a medical assessment before beginning an exercise program creates an image of exercise as potentially harmful [88].

Falls causing injury are a concern in older people, and often the reason for referral to exercise programs to address low strength and power, poor balance, poor gait and functional ability, and fear of falling. To reduce falls, the activity should include training in balance, strength, coordination and reaction times, but to reduce fractures, weight-resisted exercise is necessary [89,90]. Trials have shown that exercise programs can reduce the risk of falls and additional benefits may occur from a vision check and treatment intervention, and a home hazard check and advice intervention [91]. If older people have a history of falls, referral to specialist health professionals such as a physiotherapist, occupational therapist or experienced exercise instructors qualified in working with older people with a history of falls is recommended, and the wearing of hip protectors during exercise may be recommended [92].

Although physical activity helps in the prevention and treatment of cardiovascular disease, there are concerns about myocardial infarction and cardiac arrest associated with exercise in older people, especially those with known pre-existing heart disease. However, it is difficult to predict people in whom this might arise, as ischaemic symptoms are usually absent in people prior to any morbid event, and exercise testing has not predicted activity-related coronary events [93]. Coronary infarction is now understood as arising from plaque rupture in the coronary arteries, with one trigger for plaque rupture being exercise. This risk increases only briefly during the bout of exercise and, for those who exercise regularly, is followed by a prolonged reduction of risk between bouts [88,94]. Given the very low rate of additional risk which occurs while people are exercising, ready access to a defibrillator and ensuring that program staff are trained in how to use this and maintain cardiopulmonary resuscitation is recommended.

Professional associations, governments, peak bodies and non-government organisations often have advice for those running and participants in exercise programs for older adults, as is described in Zaleski et al. [95]. While the management of risk is very important, continuation in exercise programs depend upon participants’ experience of social and physical benefits, and on social and family support to encourage their attendance and attitudes to exercise [96].

## 5. Recommendations: What Does This Mean in Practice?

While exercise can be safe and effective, even in patients with diseases and multimorbidity, older adults have diminished physiological reserve and the ability to thermoregulate in high ambient heat as well as generally having many comorbidities. Hence, those developing physical activity programs for older adults living in rural and remote areas that experience high temperatures must consider many factors, including 1. patient factors, including physiology, pathophysiology, motivation; 2. the environment (natural and built); 3. economics (personal and government); 4. climate; and 5. availability of service providers. Consideration of these issues is relevant for all older adults, regardless of their biological age and whether they are “healthy”, although programs will need to be modified to consider the health conditions and preferences of an individual [32]. Those involved in the planning and delivery of a program need to tailor the program to take account of the relevant factors and participants, mindful of the science of thermoregulation and how it is impacted by ageing as well as local factors, participants and facilities. This is shown conceptually in Figure 1, although it is not possible to fully capture the interactions between the multiple factors in a static graphic.

Co-morbidities, motivation and understandings regarding physical activity, and their willingness and preferences for exercise will influence a person’s desire and willingness to be physical active and their options for exercising safely; this will also be affected by sociocultural factors. The current health status of the individuals could affect their participation in any program and health professional assessment may be needed. Ideally, programs should be flexible to allow maximum participation and inclusion, while also ensuring patient safety. The programs should be augmented with appropriate health messaging and health promotion activities to increase patients’ knowledge of the importance of physical activity and increase motivation. The educational program should be aligned with healthy eating guidelines and include how individuals can increase their level of incidental physical activity at home, as well as more structured and planned leisure time physical activity.

The physical and built environment both need consideration in the development of any physical activity program and opportunities will vary by location. Assessment of the relevant local physical and built environment will help identify potential barriers and opportunities to participation. Rural and remote towns will vary considerably in the walkability (footpaths and trails) in the area for older people, and factors like personal security, lighting, availability of open spaces such as ovals, parks, courtyards, beaches, and bush and the availability of sports facilities such as swimming pools, gyms, halls, and courts [70,97]. Ensuring adequate availability of cold water and regular breaks to encourage that it is being drunk and allow opportunities for skin cooling is essential to minimize the risk of dehydration and heat stress.

Understanding the climate profile (temperature, humidity, and UV exposure) of the town/region will be useful in planning for physical activity [74]. Being in a hot climate inevitably requires consideration of both the built and natural environment to determine suitability for physical activity programs for older people. The availability and use of a cool environment for participation during periods of extreme heat is recommended, with suitable options including swimming pools (indoor or outdoor), shaded areas (such as swimming pools, parks, courtyards), air-conditioned halls/gym spaces, and outdoor areas that are shaded with fans, water sprays/mist. Built environments which need ongoing maintenance or energy to maintain a comfortable temperature may be desirable, but also have potential drawbacks related to costs.

The cost to participants is a key consideration, particularly given that higher rates of chronic disease and premature mortality occur in people with less resources. Programs may be subsidized or provided at low cost. Given the costs associated with chronic disease and the many documented health benefits of participation in physical activity, governments should be encouraged to continue to invest in resources and environments to promote healthy lifestyles for older people. Advocacy of the importance of appropriate facilities, equipment, and subsidies for older adults to participate in physical activity programs needs to be directed at all levels of government. If programs are held in sports facilities such as swimming pools, there is generally a cost, and this is likely to be an impediment to inclusion and regular participation for some [98]. Opportunities for older adults to participate in physical activity that are of low or no cost should be encouraged. Transport to support older people to be able to get to and from planned structured physical activity programs is another important consideration [62].

Important sociocultural considerations impact people’s participation in physical activity, so understanding the cultural and local wishes of the community underpins successful programs. Developers should use the principles of co-design when developing community physical activity programs. Community consultation will contribute to the design and implementation of a program and increase suitability for local people and environment [99,100]. There are also benefits for those developing physical activity programs in collaborating with other health and social service providers in the community. This can facilitate the integration of physical activity program with the health management plans for individuals and communities, and the assessment and management of risks.

## 6. Conclusions

This paper brings together scientific evidence related to older age, heat and physical activity. Given the wide range of circumstances in which such programs might be planned and implemented, we have not provided specific advice for programming. This may be regarded as a limitation, but we highlight multiple considerations for planning that can encourage participation and reduce risk. The standard advice for designing exercise programs for the elderly will need to be adapted for hot climates. Critical considerations are the natural and built environment, nature and timing of exercise, plus additional considerations to reduce risk to participants through maintaining good hydration and limiting intense exercise which will induce heat production. Ensuring moderate exercise in an ambient environment where heat transfer mechanisms can occur to the external environment will help ensure participant safety and comfort. Social elements associated with physical activity programs are an important contributor to participant enjoyment and retention. Careful evaluation of relevant programs operating at this intersection of knowledge areas could help to build evidence based upon program implementation in the real world.

## Figures and Tables

**Figure 1 ijerph-18-01331-f001:**
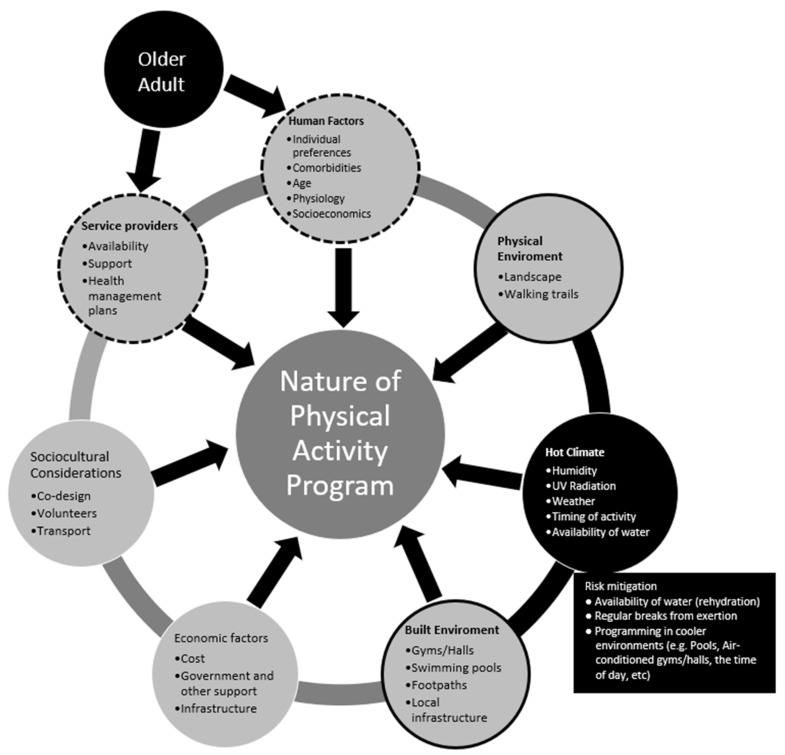
Conceptual Summary of how a Hot Environment Influences Planning Considerations for Physical Activity Programs for Older Adults.

**Table 1 ijerph-18-01331-t001:** Types of Exercises.

Type	Minimum Recommended Duration	Examples
Endurance Activities that increase heart and breathing rate. They target the heart, lungs and circulatory system.	150 min a week 30 min most days of the week NB: these can be done in short sessions ie 3 × 10 min or 2 × 15 min	Brisk walking or jogging Yard work (mowing, raking) Dancing Swimming Biking Climbing stairs or hills Playing tennis Washing the car Walking the dog Mopping and vacuuming
Strength Activities that help strengthen muscle and bone to maintain daily functions and balance and reduce falls risk.	2–3 sessions a week, separated by 1 day 2–3 sets of muscle groups, 10–12 repetitions per session NB: avoid two consecutive sessions working on the same muscle group	Weight, strength or resistance training exercises Lifting and carrying (e.g., groceries or small children) Climbing stairs Moderate yard work (e.g., digging, shifting soil) Calisthenics (push ups and sit ups)
Flexibility Activities that help to increase mobility and maintain ease of movement.	2–3 times a week (preferably everyday) For each stretch: Hold for at least 10–30 s 3–4 times each stretch	Gentle reaching, bending and reaching Tai Chi Bowls Mopping, vacuuming Yoga Dancing Gardening
Balance Activities that improve balance and stability and help prevent falls.	Everyday	Tai Chi Balance exercises Incorporate into lifestyle, e.g., balance exercise while standing in line, performing other tasks. Environmental safety important.

Note: Developed from National Institute on Ageing [28], Australian government: “Choose Health: Be active” [11] and National Physical Activity Recommendations for Older Australians [10].

**Table 2 ijerph-18-01331-t002:** Factors associated with physical inactivity and their potential modifiability.

Factors Associated with Physical Inactivity [51]	Physical Literacy Factor [29] (Intrapersonal, Interpersonal, Organisational, Community, Policy)	Ease of Modifiability (Modifiable/Not Modifiable)
**Demographic factors**		
Older age	Intrapersonal	Not modifiable
Female Sex	Intrapersonal	Not modifiable
Non-white race/ethnicity	Community	Not modifiable
Low socioeconomic status	Community	Modifiable *
**Health-related clinical factors**		
Chronic illness	Intrapersonal	Modifiable *
Poor general health and physical function	Intrapersonal	Modifiable *
Overweight/obesity	Intrapersonal	Modifiable *
**Cognitive and psychological factors**		
Greater perceived barriers to physical activity	Intrapersonal	Modifiable
Lack of enjoyment of physical activity	Intrapersonal	Modifiable
Low expectations of benefits from physical activity	Intrapersonal	Modifiable
Poor psychological health	Intrapersonal	Modifiable *
Low self-efficacy for physical activity	Intrapersonal	Modifiable *
Low self-motivation for physical activity	Intrapersonal	Modifiable *
Lack of readiness to change physical activity behaviours	Intrapersonal	Modifiable
Poor fitness level	Intrapersonal	Modifiable
**Behavioural factors**		
Prior physical activity	Intrapersonal	Modifiable
Smoking	Intrapersonal	Modifiable
Type A Behaviour (assoc’d with poorer adherence but greater overall physical activity levels)	Intrapersonal	Modifiable *
**Social factors**		
Lack of cohesion in exercise group	Interpersonal	Modifiable
Lack of physician influence/advice for physical activity	Interpersonal	Modifiable
Lack of social support for physical activity	Community	Modifiable
**Program-related factors**		
High physical activity intensity	Organisational	Modifiable
Long physical activity duration	Organisational	Modifiable
**Environmental factors**		
Lack of access to facilities/parks/trails	Community/Policy	Modifiable *
Lack of neighbourhood safety	Community/Policy	Modifiable *

Note 1: Taken from Rivera-Torres et al. [51], Adapted from Jones et al. [29]. Note 2: * Denotes ‘potentially modifiable’. Modifiability has been defined as whether changes can occur to allow for increased physical activity. Potential modifiable factors are considered as those where change is possible but would take a longer time and/or may require individual behaviour change or action from community, organisations or government policies for change, which favour physical activity.

## Data Availability

Not applicable.

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
