# Peer review of "Considerations in Planning Physical Activity for Older Adults in Hot Climates: A Narrative Review"

_ijerph, 2021, doi:10.3390/ijerph18031331_

Round 1
Reviewer 1 Report
Thanks for responding to my concerns and comments. I am fully satisfied with the responses and changes made by the authors.
Reviewer 2 Report
Dear authors,
My considerations were directed at the first version of the manuscript. Perhaps that is why you misunderstood them (e. g., specific population). However, I invite the authors to look at the initial paper they submitted to the journal and compare it with the latest version submitted. Undoubtedly, a substantial improvement of work has been achieved. I still have slight doubts about the narrative revisions. However, the final decision will be up to the editor. Congratulations on the progression from the first version submitted and this latest one.
Only a little question. Please, modify the references to the style of the IJERPH journal.
Best wishes,
Reviewer 3 Report
Congratulations. I believe that your work has improved considerably.
This manuscript is a resubmission of an earlier submission. The following is a list of the peer review reports and author responses from that submission.
Round 1
Reviewer 1 Report
Thanks for submitting the manuscript to IJERPH. The authors provided a narrative review to informs planning of physical activity programs for older people living in hot climates. The review was comprehensive and well written. However, there are several minor issues need to be addressed by the authors:
1) Please provide the latest figures (p. 3, line 44).
2) On p. 3 (lines 50-74), the authors mainly highlighted the situations in Australia, UK and US. The situation in UK may not be relevant, as most of the territories in UK with moderate climate. The authors may consider to engage with the literature more related to the areas with hot climates.
3) The authors claim that ‘This narrative review was undertaken to inform planning for a physical activity program to be delivered in a regional city in the Pilbara region of WA, at latitude 21-22° South’ (p. 3, lines 81-82). To what extent the findings of this study are applicable to other regions in the world with hot climate? The research findings may hardly reach out to the wider audiences if the current study with limited focus.
4) The section ‘3. Definitions’ should be restructure it into something like framework or analytical framework with justifications from the relevant literature/theory.
5) Please provide the sources/references on p. 15 (lines 401-412).
6) The authors stated that ‘Professional associations, governments, peak bodies and non-government organisations often have advice for those running and participants in exercise programs for older adults, as is described in Zaleski et al’ (p. 16, lines 440-442). The authors may need to further highlight the novelty and contribution of this study.
7) In the recommendations (p. 16), to what extent the recommended practices are relevant to older adults who are healthy and with medical conditions?
8) Minor issues: Need to introduce the abbreviation before using it, e.g. PA (p. 5, line 141); The table numbers are not correct. The authors used Table 2 on p. 5 and then Table 1 on p. 8.
Author Response
Reviewer1 – PA and older people
Thanks for submitting the manuscript to IJERPH. The authors provided a narrative review to informs planning of physical activity programs for older people living in hot climates. The review was comprehensive and well written. However, there are several minor issues need to be addressed by the authors:
- Please provide the latest figures (p. 3, line 44).
We have quoted a 2019 primary reference from a pre-eminent author (Lopez) who looks at life expectancy in Australia over time. This is an up to date reference.
- On p. 3 (lines 50-74), the authors mainly highlighted the situations in Australia, UK and US. The situation in UK may not be relevant, as most of the territories in UK with moderate climate. The authors may consider to engage with the literature more related to the areas with hot climates.
We agree with the reviewer’s comment that the UK does not generally experience a very high hot climate. However, this is in the introduction which frames the relevance of the article more broadly - we were making the general point about many countries having ageing populations which fail to meet physical activity guidelines. At this point in the article, we have not even raised the issues of climate and heat which are elaborated on in the following paragraphs.
- The authors claim that ‘This narrative review was undertaken to inform planning for a physical activity program to be delivered in a regional city in the Pilbara region of WA, at latitude 21-22° South’ (p. 3, lines 81-82). To what extent the findings of this study are applicable to other regions in the world with hot climate? The research findings may hardly reach out to the wider audiences if the current study with limited focus.
Thank you for raising this. The review does have much wider relevance than just the specific situation which resulted in our interest int his topic. We have now made a change to make the broader relevance clear “As similar conditions occur across northern Australia and in many other areas of the world, the review has much broader relevance. Moreover, global warming means that extreme weather events and heat stress are likely to increase [12].”
- The section ‘3. Definitions’ should be restructure it into something like framework or analytical framework with justifications from the relevant literature/theory.
We do not believe this warrants a framework or analytical framework. We tried to look at how this section could be incorporated into the introduction but felt this interrupted the introduction which is primarily to frame the purpose of the review. This section on definitions was included so that readers who do not have an extensive knowledge of relevant literature (the relevant readership for a narrative review) would understand the relevant terminology. We have chosen to retain most elements of this section but have removed the section on physical literacy which is not germane to the topic of the review.
- Please provide the sources/references on p. 15 (lines 401-412).
Apologies, the referencing to this section has been added
6) The authors stated that ‘Professional associations, governments, peak bodies and non-government organisations often have advice for those running and participants in exercise programs for older adults, as is described in Zaleski et al’ (p. 16, lines 440-442). The authors may need to further highlight the novelty and contribution of this study.
Zaleski et al wrote an article which looked at the prescription of exercise in older adults. It does not consider heat, or hot climates but examines the impact when prescribing exercise for specific chronic diseases. This is only one aspect considered in this scoping review which explores a much wider range of relevant issues for the context of older people, physical activity and hot climates.
- In the recommendations (p. 16), to what extent the recommended practices are relevant to older adults who are healthy and with medical conditions?
We have added an additional sentence to highlight the salience of the identified areas for consideration in exercise planning for all older adults, regardless of their health and the need to tailor programs to individuals
- Minor issues: Need to introduce the abbreviation before using it, e.g. PA (p. 5, line 141); The table numbers are not correct. The authors used Table 2 on p. 5 and then Table 1 on p. 8.
Our apologies for these minor errors. We decided against using PA as an abbreviation for physical activity, but this one instance slipped through and we have now replaced the abbreviation with the words. We renumbered the Tables correctly but have now decided to remove the table on physical literacy.

Reviewer 2 Report
Thank you for inviting me to review this article. This review tried to inform planning of physical activity programs for older people 15 living in rural areas with very hot climates for a period of the year. This review could certainly be of interest to the target study population. However, since it is a narrative (non-systematic) review and since it is addressed to a population with such specific characteristics, I cannot accept this manuscript.
Best wishes,
Author Response
Thank you for inviting me to review this article. This review tried to inform planning of physical activity programs for older people 15 living in rural areas with very hot climates for a period of the year. This review could certainly be of interest to the target study population. However, since it is a narrative (non-systematic) review and since it is addressed to a population with such specific characteristics, I cannot accept this manuscript.
Thanks for your comments.
Narrative reviews are reported to “constitute the largest share of all text types in medicine and they concluded that they “remain the staple of medical literature” [1]. Narrative literature review articles have an important role in continuing education because they provide readers with up-to-date knowledge about a specific topic or theme. They do not attempt to do what systematic review do, that is enable reproduction of data or answer specific quantitative research questions. The reviewer could be directed to read the paper by Greenhalgh and colleagues which makes an excellent case for narrative reviews [2]. Even though it should not be needed, we have now cited this paper as part of the justification of this approach.
We have revised the manuscript to make it clearer that while our interest arose because of specific project localised to an older population in a hot climate, that the review has much wider relevance to other hot climates where older people live.
- Bastian H, Glasziou P, Chalmers I. Seventy-five trials and eleven systematic reviews a day: how will we ever keep up? PLoS Med. 2010;7(9):e1000326. https://doi.org/10.1371/journal.pmed.1000326.
- Greenhalgh T, Thorne S, Malterud K. Time to challenge the spurious hierarchy of systematic over narrative reviews? Eur J Clin Invest. 2018 Jun;48(6):e12931. doi: 10.1111/eci.12931. Epub 2018 Apr 16.
Reviewer 3 Report
Dear Editor,
The work presented consists of a narrative review that aims to provide aspects of the design of physical activity in adults. The manuscript is well written and the quality is good.
Although the topic addressed is of interest, I consider that it is not a suitable subject for the journal for several reasons:
1) It is a narrative review
2) It is a very specific topic
3) Authors provide very few practical aspects of the research question
4) There is an imbalance between the introductory part and the topic-specific part of the review.
Author Response
The work presented consists of a narrative review that aims to provide aspects of the design of physical activity in adults. The manuscript is well written and the quality is good.
Although the topic addressed is of interest, I consider that it is not a suitable subject for the journal for several reasons:
1) It is a narrative review
2) It is a very specific topic
3) Authors provide very few practical aspects of the research question
4) There is an imbalance between the introductory part and the topic-specific part of the review.
Thanks for your comments.
1 & 2.
This is a narrative review which is a well described and common methods for bringing together and discussing the state of the science on a specific topic or theme from a theoretical and contextual point of view. It does indeed address a very specific topic but one which is informed by a wide range of scientific literature to accommodate the many factors that impinge on safe and effective planning for physical activity programs in an important and growing population group.
3 & 4 – suggestions for what we should do to address these comments?
We presume that the reviewer was seeking more practical recommendations here? We had thought this might be possible but ultimately felt that we are providing a review on a broad topic which intersects physical activity, older age and hot climate. We had titled our paper “Considerations in planning…” and do not think that given the large range of circumstances, it is possible to provide definitive program recommendations. We have added Figure 1 which we hope makes more clearly evident that physical and built environment are critically important factors which influence the advice on physical activity programs for older people in high ambient temperatures.
Reviewer 4 Report
Thank you very much for considering me as a reviewer of this study. In my opinion, much of the future research is focused on older adults, as the demographic evolution of the most developed countries in the world is clearly an ageing population. Physical activity and exercise should be part of the daily habits of the whole population, and special attention should be paid when it is done in very cold or hot climates like this one.
This focus on Australia can clearly be extrapolated to other countries, especially from the Mediterranean climate such as Italy or Spain, which also have very hot weather conditions in summer. Just like Australia, these countries also have a high life expectancy of their population due to their own climate and living conditions.
Having said this, I must make some observations about the work, which I consider very important to be able to establish the basic lines of action in physical exercise for this age group.
Firstly, the authors mark too many keywords, duplicating many of those that appear in the title, such as "physical activity", "older adults", or "climate". Environment and environment are similar words so I would leave only one.
Introduction
The introduction in general seems to me to be adequate. However, the introduction does not deal with the work of physical activity in hot environments (in the elderly or normal adults), which allows us to go deeper into the relevance of the subject.
Likewise, some comments made by the authors could be supported by some previous reference such as the text in lines 41-42, lines 57-58, or lines 190-193.
The authors in lines 42-46 talk about the increase of the older adult population and its life expectancy. It would be advisable to include some future estimate of how far the proportion of older adults can reach in the coming years, surely some document or study makes an estimate of the demographic evolution.
Finally, it would also be interesting to be able to talk about the current situation in the Pilbara region, which is where the study is focused, about what population it has, what proportion is older adults, etc.
Method
I must point out that the method seems to me to be very scarce and insufficient, and needs to be expanded and made much more specific.
Why haven't you searched databases such as Web of Science or Scopus? They include more journals that can include relevant studies.
What is the period in which the search was carried out? What results were obtained? How many studies were selected? Are there inclusion or exclusion criteria? How many researchers carried out the search and what previous experience did they have?
What procedure was carried out? It is important to develop a section that allows other researchers to replicate this study.
Section 3
Section 3 "Definitions" would remove it, including the definitions and information on the different concepts within the introduction. For example, address physical activity and exercise when talking about physical activity and then address older adults.
Regarding the Tables, I do not understand why Table 1 comes after Table 2. The spaces in Table 2 could be adjusted to fit all on one sheet. Moreover, it is necessary to reference where all the information appears in it since neither in the text when it is mentioned, nor in the table itself, there are references as if it happened in Table 1. Line 177 is missing a point after "al".
Section 4
In sports performance there is a wide field of study on heat work, perhaps information should also be included on how it affects the general population or risk groups, as I believe that too much work is being quoted in this section by Balmain et al [30]. References should be expanded and could be more varied.
The same is true of the over-quoted subsection on 'Barriers and Enablers', which is based on the study by Bethancourt et al.
Table 3 should be "Table 3. With respect to ease of movement, the authors explain and clarify that it is modifiable. However, what criteria have been established to determine whether and to what extent this factor can be modified?
Section 5
The authors make practical considerations about narrative review. However, in the title they present "Planning physical activity". It is necessary to include more specific recommendations about what type of physical activity to do in hot conditions, its frequency and duration, as well as to be able to summarize the most relevant factors or aspects to carry out this physical activity or exercise (a table or figure can be included for this purpose).
Finally, the authors should consider including a section on limitations of their study and commenting on future lines of research that should be addressed on this subject in order to improve knowledge in a more empirical way.
References
The references should be reviewed as they do not comply with the journal's standards. They need to be justified. For example, some references include the full name of the journal and not its abbreviated form.
Author Response
Thank you very much for considering me as a reviewer of this study. In my opinion, much of the future research is focused on older adults, as the demographic evolution of the most developed countries in the world is clearly an ageing population. Physical activity and exercise should be part of the daily habits of the whole population, and special attention should be paid when it is done in very cold or hot climates like this one.
This focus on Australia can clearly be extrapolated to other countries, especially from the Mediterranean climate such as Italy or Spain, which also have very hot weather conditions in summer. Just like Australia, these countries also have a high life expectancy of their population due to their own climate and living conditions.
We agree and have added more on this
Having said this, I must make some observations about the work, which I consider very important to be able to establish the basic lines of action in physical exercise for this age group.
Firstly, the authors mark too many keywords, duplicating many of those that appear in the title, such as "physical activity", "older adults", or "climate". Environment and environment are similar words so I would leave only one.
The journal states that up to 10 keywords may be used. We have removed three keywords that were not very specific .
Introduction
The introduction in general seems to me to be adequate. However, the introduction does not deal with the work of physical activity in hot environments (in the elderly or normal adults), which allows us to go deeper into the relevance of the subject.
The introduction is to primarily frame the topic of the review. We have added a sentence which describes that heat is generated as part of physical activity but since there is a whole section on thermoregulation as part of the review have not elaborated further at this point.
Likewise, some comments made by the authors could be supported by some previous reference such as the text in lines 41-42, lines 57-58, or lines 190-193.
We consider that lines 41-42 are already adequately referenced. We have added the references to (what was) lines 57-58 and references to lines 190-193.
The authors in lines 42-46 talk about the increase of the older adult population and its life expectancy. It would be advisable to include some future estimate of how far the proportion of older adults can reach in the coming years, surely some document or study makes an estimate of the demographic evolution.
We have added more information on demographic ageing as a global issue with increasing life expectancy and the encouragement of exercise in older adults being widely adopted to efforts to reduce premature disability through health behaviour change that includes exercise at older ages.
Finally, it would also be interesting to be able to talk about the current situation in the Pilbara region, which is where the study is focused, about what population it has, what proportion is older adults, etc.
Relevance of the Pilbara was only to piquing our interest in the issue given our work there. The review has much broader relevance to exercise programs for older people in hot climates. We have reduced the emphasis on this specific location.
Method
I must point out that the method seems to me to be very scarce and insufficient and needs to be expanded and made much more specific.
Why haven't you searched databases such as Web of Science or Scopus? They include more journals that can include relevant studies.
This is not a systematic review and so does not require the identification of all relevant studies, but rather a summary of salient issues for the question to be explored. We have added further detail on our approach to the review and methodology.
What is the period in which the search was carried out? What results were obtained? How many studies were selected? Are there inclusion or exclusion criteria? How many researchers carried out the search and what previous experience did they have?
What procedure was carried out? It is important to develop a section that allows other researchers to replicate this study.
This is a narrative review and not a systematic review for which these questions would be relevant. The review was undertaken in 2020. We have added this information to the methods.
Section 3
Section 3 "Definitions" would remove it, including the definitions and information on the different concepts within the introduction. For example, address physical activity and exercise when talking about physical activity and then address older adults.
Given this is a narrative review for people wanting to get an overall understanding of the field, we believe it is important to retain definitions of physical activity, types of exercise and what we mean by the term older people. We have removed the section on physical literacy.
Regarding the Tables, I do not understand why Table 1 comes after Table 2. The spaces in Table 2 could be adjusted to fit all on one sheet. Moreover, it is necessary to reference where all the information appears in it since neither in the text when it is mentioned, nor in the table itself, there are references as if it happened in Table 1. Line 177 is missing a point after "al".
Apologies for the wrong numbering of Tables 1 and 2. Table 1 which had al without a full stop has been removed and the tables renumbered in the correct order.
Section 4
In sports performance there is a wide field of study on heat work, perhaps information should also be included on how it affects the general population or risk groups, as I believe that too much work is being quoted in this section by Balmain et al [30]. References should be expanded and could be more varied.
We have expanded upon the references in this section
The same is true of the over-quoted subsection on 'Barriers and Enablers', which is based on the study by Bethancourt et al.
We have considerably expanded upon the relevant discussion and referencing in this section
Table 3 should be "Table 3. With respect to ease of movement, the authors explain and clarify that it is modifiable. However, what criteria have been established to determine whether and to what extent this factor can be modified?
We have included more information on modifiability and how this was conceived in the text (it was previously primarily in the footnote).
Section 5
The authors make practical considerations about narrative review. However, in the title they present "Planning physical activity". It is necessary to include more specific recommendations about what type of physical activity to do in hot conditions, its frequency and duration, as well as to be able to summarize the most relevant factors or aspects to carry out this physical activity or exercise (a table or figure can be included for this purpose).
We have now added a figure to summarise the information on considerations in planning which is the title of this article. Given the wide range of circumstances, we have not made specific recommendations.
Finally, the authors should consider including a section on limitations of their study and commenting on future lines of research that should be addressed on this subject in order to improve knowledge in a more empirical way.
We have added a comment to the conclusion about the lack of specific programming recommendations and this can be regarded as a limitation.
References
The references should be reviewed as they do not comply with the journal's standards. They need to be justified. For example, some references include the full name of the journal and not its abbreviated form.
We have used the MDPI Endnote referencing style. We have previously advised the journal that there are problems with this style. We have fixed some minor errors of referencing in Endnote and will ensure that the references are properly formatted for the journal’s standards once the article is finalised and the references can be delinked.

Round 2
Reviewer 2 Report
My biggest concerns about this article were directed at the very specific population identified (in addition to the lack of practical applications). The full explanation of the rejection of the article was sent to the editor at the time. Because the article was rejected in the first instance, no changes were indicated. As a researcher, I know perfectly well what a narrative review and a systematic review are. The opinion of some researchers, even if it is published, does not make that thought an absolute truth. Although narrative reviews have their interest and importance, what is new about this review? What do the authors propose (practical applications) so that the older adults can engage in physical activity in hot climates? What about the new recommendations of the World Health Organization recently published?
Honestly, the article has improved remarkably (with the change of target population) and is well written. However, in my humble opinion, it has no place in such a high impact journal as IJERPH.
Best wishes,
Reviewer 3 Report
The manuscript has been considerably improved in its present form.
The responses have satisfied this reviewer.
The introduction of the figure allows a better understanding of the manuscript.
I therefore suggest that it be accepted.
Reviewer 4 Report
Merry Christmas, I hope the authors and their families are well at this time of year. In reviewing your document again, I must make a number of small observations:
1. I repeat my insistence on modifying the keywords as it is not advisable to include terms that already appear in the title. It reduces your visibility in the search if you only use the same terms and do not take advantage of other synonyms. Key words such as "physical activity" or "older adults" would be removed and replaced by, for example, "sport and elderly", which are synonymous terms and would apply to the visibility of the document.
2. I am clear about the differences between a systematic and a narrative review. However, I must point out that "Google Scholar" is not a database as such as it does not carry out any kind of scientific filtering. My previous comments are along the lines that although PUBMED and CINAHL are used as specialised databases, others such as WOS or Scopus include other types of journals that could have had more relevant information.
3. I do not agree with the authors in their assessment that this is an introductory article and therefore it is necessary to leave a section with terminological clarification. I believe that people who can show interest will already have some basic knowledge about what physical activity is, which can be integrated into the introduction as I commented in my previous review.
4. In the search strategy, it would be interesting to be able to specify what type of documents have been contemplated, whether articles, books, book chapters, contributions at conferences, grey literature, etc.